# Prevalence and impact of chronic dysglycaemia among patients with COVID-19 in Swedish intensive care units: a multicentre, retrospective cohort study

Anca Balintescu  ,[1] Susanne Rysz,[2] Carl Hertz,[3] Jonathan Grip,[2] Maria Cronhjort,[1] Anders Oldner,[2] Christer Svensen,[1] Johan Mårtensson[2]

¹Department of Clinical Science and Education, Södersjukhuset, Karolinska Institute, Stockholm, Sweden
²Department of Perioperative Medicine and Intensive Care, Karolinska Institute, Stockholm, Sweden
³Department of Anesthesia and Intensive Care, Stockholm South General Hospital Anaesthesia, Stockholm, Sweden

**Correspondence to**
Dr Anca Balintescu;
anca.balintescu@ki.se

## ABSTRACT

**Objective** Using glycated haemoglobin A1c (HbA1c) screening, we aimed to determine the prevalence of chronic dysglycaemia among patients with COVID-19 admitted to the intensive care unit (ICU). Additionally, we aimed to explore the association between chronic dysglycaemia and clinical outcomes related to ICU stay.
**Design** Multicentre retrospective observational study.
**Setting** ICUs in three hospitals in Stockholm, Sweden.
**Participants** COVID-19 patients admitted to the ICU between 5 March 2020 and 13 August 2020 with available HbA1c at admission. Chronic dysglycaemia was determined based on previous diabetes history and HbA1c.
**Primary and secondary outcomes** Primary outcome was the actual prevalence of chronic dysglycaemia (pre-diabetes, unknown diabetes or known diabetes) among COVID-19 patients. Secondary outcome was the association of chronic dysglycaemia with 90-day mortality, ICU length of stay, duration of invasive mechanical ventilation (IMV) and renal replacement therapy (RRT), accounting for treatment selection bias.
**Results** A total of 308 patients with available admission HbA1c were included. Chronic dysglycaemia prevalence assessment was restricted to 206 patients admitted ICUs in which HbA1c was measured on all admitted patients. Chronic dysglycaemia was present in 82.0% (95% CI 76.1% to 87.0%) of patients, with pre-diabetes present in 40.2% (95% CI 33.5% to 47.3%), unknown diabetes in 20.9% (95% CI 15.5% to 27.1%), well-controlled diabetes in 7.8% (95% CI 4.5% to 12.3%) and uncontrolled diabetes in 13.1% (95% CI 8.8% to 18.5%). All patients with available HbA1c were included for the analysis of the relationship between chronic dysglycaemia and secondary outcomes. We found no independent association between chronic dysglycaemia and 90-day mortality, ICU length of stay or duration of IMV. After excluding patients with specific treatment limitations, no association between chronic dysglycaemia and RRT use was observed.
**Conclusions** In our cohort of critically ill COVID-19 patients, the prevalence of chronic dysglycaemia was 82%. We found no robust associations between chronic dysglycaemia and clinical outcomes when accounting for treatment limitations.

## STRENGTHS AND LIMITATIONS OF THIS STUDY

⇒ It presents the prevalence of chronic dysglycaemia in an intensive care unit (ICU) population with COVID-19 based on additional quantification of admission haemoglobin A1c.
⇒ Actual prevalence of chronic dysglycaemia calculation in all ICU admitted patients, reducing the risk of ascertainment bias.
⇒ Treatment limitations were considered in the analysis of clinical outcomes, thereby reducing the risk of treatment selection bias.
⇒ We lack data on glycaemic control during ICU stay that might have influenced clinical outcomes.

## BACKGROUND

Diabetes has been identified as a frequent comorbidity in patients with COVID-19, with a prevalence ranging from 7.4% to 34.3% among those requiring hospitalisation.[1–3] A meta-analysis published in April 2020 found diabetes to be the second most frequent comorbidity in patients with COVID-19 admitted to the intensive care unit (ICU).[4] Furthermore, COVID-19 patients with diabetes appear to have a significantly higher risk of ICU admission and worse prognosis than COVID-19 patients without diabetes.[4–6] Particularly, a glycated haemoglobin A1c (HbA1c) level above 7% (53 mmol/mol) was identified as a risk factor for ICU admission.[7]

Recent data also indicate that diabetes is associated with worse prognosis among ICU patients with COVID-19.[8] However, these studies did not include HbA1c measurements to identify patients with pre-diabetes or previously undiagnosed diabetes. This is an important limitation since both pre-diabetes and diabetes is considerably underdiagnosed both in the community[9] and in the ICU.[10]

Additionally, a history of diabetes diagnosis and HbA1c at ICU admission, measured in consecutively admitted patients, is important in determining the true prevalence of chronic dysglycaemia in the critically ill COVID-19 population. Finally, information about limitations of life-sustaining treatment was not considered in previous outcome analyses. This is unfortunate since the presence of such limitations may introduce treatment selection bias.

We, therefore, conducted a multicentre observational study using quantification of HbA1c and information about diabetes history to determine the actual prevalence of chronic dysglycaemia (pre-diabetes, unknown diabetes or known diabetes) among COVID-19 patients admitted to ICU. In addition, we aimed to explore the relationship of chronic dysglycaemia with 90-day mortality, ICU length of stay, duration of invasive mechanical ventilation (IMV) and severe acute kidney injury requiring renal replacement therapy (RRT) accounting for treatment selection bias. We hypothesised that the prevalence of chronic dysglycaemia in COVID-19 patients admitted to the ICU exceeds the prevalence of chronic dysglycaemia in the non COVID-19 critically ill population. Moreover, we hypothesised that such chronic dysglycaemia would be associated with worse clinical outcomes during ICU stay in patients with COVID-19.

## MATERIAL AND METHODS

The study was performed in accordance with the Helsinki Declaration and reported in conformity with the Strengthening the reporting of observational studies in epidemiology (STROBE statement.[11]

### Patient and public involvement statement

The study is based on data that was collected during the ongoing COVID-19 pandemic in a quality register. No intervention was applied to the individual patient. The public and patients were not involved in the design of the study. Results are to be disseminated to the public and scientific community through publication in peer-reviewed journal with open access.

### Study design

We conducted a multicentre, retrospective observational study of adult (≥18 years) patients with a positive PCR for SARS-CoV-2 admitted to 10 ICUs in 3 hospitals in Stockholm, Sweden between 5 March 2020 and 13 August 2020 (first wave). We excluded patients without HbA1c obtained on admission to the ICU, patients in the third trimester of pregnancy and patients with a primary admission diagnosis other than COVID-19. In patients with multiple ICU admissions, only the first admission was considered. All included patients were assessed in the outcome analyses. Assessment of chronic dysglycaemia prevalence was restricted to a nested cohort of patients from ICUs in which HbA1c measurement was included in the routine laboratory panel performed on all consecutive admissions. In the prevalence analysis, we, therefore, excluded patients with available HbA1c who were admitted to ICUs in which HbA1c was measured only at the discretion of the treating clinicians.

### Data collection

HbA1c was measured in whole blood at ICU admission using the VARIANT II TURBO Haemoglobin Testing System analyzer (Bio-Rad Laboratories) and was reported in mmol/mol (IFCC calibrated) and in %. HbA1c was measured as part of routine care in three ICUs and at the discretion of the treating clinician in seven ICUs. We collected information on demographics, comorbidity, chronic medication, HbA1c value, mortality and decision regarding limitation of life-sustaining care from the patients' medical records (Take Care (CompuGroup Medical, Koblenz, Germany)). International Classification of Disease 10 codes were used to identify comorbidity and a history of diabetes. Additionally, data regarding known diabetes diagnosis were extracted manually from the patients' medical records. Data on body mass index, Simplified Acute Physiology Score (SAPS) 3, ICU length of stay, duration of IMV and RRT were collected from the ICU electronic patient data management system CliniSoft (GE, Barrington, Illinois, USA).

### Pre-diabetes and diabetes definitions

Pre-diabetes and diabetes were diagnosed based on two complementary methods; level of HbA1c at admission and a medical history of diabetes, and categorised into five groups:
1. No diabetes (HbA1c <42 mmol/mol (6.0%) and no history of diabetes).
2. Pre-diabetes (HbA1c 42–47 mmol/mol (6.0%–6.4%) and no history of diabetes).
3. Unknown diabetes (HbA1c ≥48 mmol/mol (6.5 %) and no history of diabetes).
4. Controlled diabetes (HbA1c<52 mmol/mol (6.9 %) and a history of diabetes).
5. Uncontrolled diabetes (HbA1c≥52 mmol/mol (6.9 %) and a history of diabetes).

In Sweden, the diagnosis of pre-diabetes and diabetes is based on the WHO's HbA1c cut-off values,[12] not the American Diabetes Association's (ADA). Therefore, we used the WHO criteria to classify the study groups in our research.

Individuals in group (2), (3), (4) and (5) were considered to have chronic dysglycaemia compared with those in group (1) labelled 'no chronic dysglycaemia'.

### Outcomes

The primary outcome was the prevalence of chronic dysglycaemia. Secondary outcomes included 90-day mortality, ICU length of stay, duration of IMV and RRT use.

### Statistical analysis

We analysed data using STATA V.12.1 (StataCorp).

Categorical data are presented as numbers and percentages and compared using the Fisher's exact test. Continuous data are summarised as median with IQR and compared using the Mann-Whitney U test. The prevalence of chronic dysglycaemia (primary outcome) was presented as percentages with 95% CIs. We displayed time to death within 90 days using Kaplan-Meier curves. Survival curves were compared using a log-rank test. We used multivariable Cox regression analysis to assess the association between chronic dysglycaemia and 90-day mortality. We used multivariable linear regression analysis to assess the association with ICU length of stay and duration of IMV. Both these outcomes were found to be well approximated by log-normal distributions and were therefore log-transformed before analysis with results presented as geometric means (95% CI). We used multivariable logistic regression analysis to assess the association with RRT use, before and after excluding patients with RRT as a treatment limitation. All regression analyses were conducted using the following models: adjusted for SAPS 3, age and sex, and adjusted for SAPS 3, age, sex, hypertension, any malignancy, any treatment limitation on admission and chronic corticosteroid use. A post hoc exploratory comparison between subgroups was done for 90-day mortality and RRT use. A two-sided $p < 0.05$ was considered statistically significant.

## RESULTS
### Patients
A total of 584 patients with positive SARS-CoV-2 test were admitted to the study ICUs during the study period. We excluded 225 patients without available HbA1c, 6 pregnant patients, 16 readmissions and 29 patients without symptoms associated with COVID-19. Therefore, we included 308 patients with available HbA1c for outcome analysis. Among those 308 patients, 206 consequently admitted patients in which HbA1c was included in the admission routine laboratory panel were used for prevalence calculation (figure 1). Baseline characteristics and treatment limitations of the entire study population are detailed in table 1.

Patients with chronic dysglycaemia were older, were more likely to have hypertension, malignancy and/or chronic kidney disease, and had higher SAPS 3 than patients without chronic dysglycaemia. Overall, 14 (22.9%) patients in the no chronic dysglycaemia group and 53 (21.5%) patients in the chronic dysglycaemia group received one or more limitations of life-supporting therapies during their ICU stay. 'No cardiopulmonary resuscitation' was the most common treatment limitation. We observed the highest proportion of limitations among patients with known (controlled or uncontrolled) diabetes. Decision to switch to palliative care was made in 1 (1.6%) patient

in the no chronic dysglycaemia group and 34 (13.8%) patients in the chronic dysglycaemia group (p=0.006). Cumulative percentage of treatment limitations relative ICU admission is displayed in online supplemental figures S1–S4.

### Primary outcome
In the nested cohort of 206 consecutive patients with available HbA1c, 169 (82.0%; 95% CI 76.1% to 87.0%) were diagnosed with chronic dysglycaemia. Pre-diabetes was present in 83 (40.2%, 95% CI 33.5% to 47.3%), unknown diabetes in 43 (20.9%, 95% CI 15.5% to 27.1%), well-controlled diabetes in 16 (7.8%, 95% CI 4.5% to 12.3%) and uncontrolled diabetes in 27 (13.1%, 95% CI 8.8% to 18.5%) patients (figure 2).

### Secondary outcomes
Nine (14.7%) patients in the no chronic dysglycaemia group and 62 (25.1%) patients in the chronic dysglycaemia group died within 90 days (p=0.09) (table 2, figure 3, online supplemental figure S5). ICU length of stay and duration of IMV were similar in the two groups. IMV was delivered to 42 (68.8%) patients without chronic dysglycaemia and to 187 (75.7%) patients with chronic dysglycaemia (p=0.32). RRT was delivered to 17 (27.9%) no chronic dysglycaemia patients and 42 (17.0%) chronic dysglycaemia patients (p=0.06) (tables 2 and 3).

On multivariable regression analysis, we observed a numerically higher mortality (adjusted HR 1.54, 95% CI 0.74 to 3.19, p=0.24) and significantly lower RRT use (adjusted OR 0.49, 95% CI 0.24 to 0.99, p=0.04) in patients with chronic dysglycaemia (table 3). No association with RRT was observed after exclusion of patients with 'no RRT' as treatment limitation. In the post hoc exploratory comparison between subgroups, RRT use was higher in the no diabetes group compared with the controlled diabetes group, as well as in the uncontrolled diabetes compared with controlled diabetes group online supplemental table S1. Individuals with uncontrolled diabetes had the lowest probability of survival followed by individuals with controlled diabetes and pre-diabetes. The highest probability of survival was observed among patients with no chronic dysglycaemia and pre-diabetes, respectively (online supplemental figure S5). However, we observed no statistically significant differences in mortality in the post hoc comparison of subgroups (online supplemental table S1).

## DISCUSSIONS
### Key findings
We performed a multicentre observational investigation to determine the prevalence of chronic dysglycaemia and its impact on clinical outcomes among COVID-19 patients admitted to ICU. Using available information about the patients' diabetic status in combination with routine

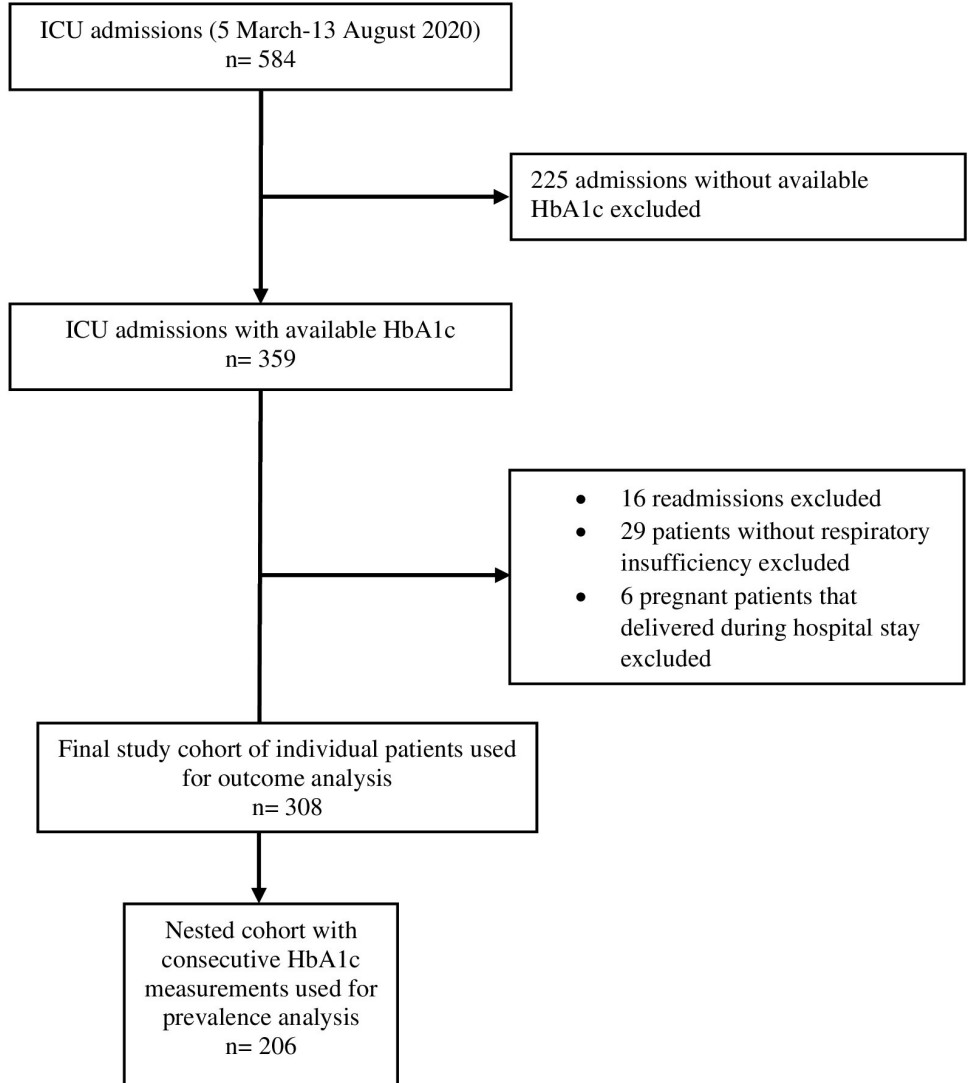

**Figure 1** Flow chart of study population. HbA1c, haemoglobin A1c; ICU, intensive care unit.

HbA1c assessment, we found that 82% had chronic dysglycaemia with two-thirds having either pre-diabetes or undiagnosed diabetes. We observed numerically higher 90-day mortality in patients with chronic dysglycaemia, with the highest mortality (31%) observed among those with uncontrolled diabetes. Conversely, the proportion of patients receiving RRT was lower among patients with chronic dysglycaemia even when patients without 'no RRT' as treatment limitation were considered separately. We found no association of chronic dysglycaemia with ICU length of stay or duration of IMV.

### Relationship with previous studies

A global meta-analysis of more than 16 000 ICU patients with COVID-19 suggests a pooled prevalence of known diabetes between 23% and 31%,[13] close to the observed prevalence in our study (21%). However, few studies have used additional HbA1c measurements to assess the actual prevalence of chronic dysglycaemia, including pre-diabetes and undiagnosed diabetes. One such ICU study from Austria found a prevalence of chronic dysglycaemia

of 85%, which is in close agreement with our findings.[14] However, the Austrian study did not assess consecutive patients and may therefore be prone to selection bias.

Our findings indicate that chronic dysglycaemia is more common in COVID-19 patients than in ICU patients with other admission diagnoses. In fact, in a pre-COVID-19 cohort of general ICU patients, we found a corresponding dysglycaemia prevalence of 33%.[10] The relationship between severe SARS-CoV-2 infection and dysglycaemia has different potential explanations. SARS-CoV-2 enters cells in various organs, including the pancreas, via ACE2. As ACE2 is involved in regulating pancreatic beta-cell function, a link between SARS-CoV-2 infection and beta-cell dysfunction and diabetes development has been suggested.[15] Interestingly, the prevalence of elevated HbA1c below the diabetes diagnostic threshold (pre-diabetes) was markedly higher in our COVID-19 cohort than in our previous pre-COVID-19 cohort (40% vs 9%). It is possible that the duration of COVID-19 symptoms before ICU admission (typically ten

**Table 1** Baseline characteristics and treatment limitations

| Characteristic | No chronic dysglycaemia | Chronic dysglycaemia | | | | P value* |
| | | Pre-diabetes | Unknown diabetes | Controlled diabetes | Uncontrolled diabetes | |
| --- | --- | --- | --- | --- | --- | --- |
| No (%) | 61 (19.8) | 114 (37.0) | 60 (19.4) | 25 (8.11) | 48 (15.5) | |
| Age, years | 57 (51, 63) | 61 (53, 68) | 60 (52, 68) | 63 (57, 71) | 62 (55, 69) | 0.03 |
| Male sex | 48 (78.6) | 92 (80.7) | 47 (78.3) | 21 (84.0) | 36 (75.0) | 1.00 |
| Body mass index†, kg/m$^2$ | 27 (25, 32) | 27 (25, 30) | 28 (25, 31) | 29 (26, 32) | 30 (26, 33) | 0.97 |
| HbA1c, mmol/mol | 39 (36, 40) | 44 (43, 46) | 51 (49, 57) | 47 (44, 49) | 70 (61, 81) | <0.001 |
| Diabetes treatment | | | | | | |
| Diet only | | | | 6 (24.0) | 2 (4.1) | |
| OAD only | | | | 17 (68) | 19 (39.5) | |
| Insulin only | | | | 1 (4.0) | 12 (25.0) | |
| OAD+insulin | | | | 1 (4.0) | 15 (31.2) | |
| Comorbidity | | | | | | |
| Hypertension | 18 (29.5) | 40 (35.0) | 23 (38.3) | 16 (64.0) | 34 (70.8) | 0.02 |
| Heart failure | 6 (9.8) | 5 (4.3) | 6 (10.0) | 0 (0.0) | 3 (6.2) | 0.24 |
| Previous myocardial infarction | 2 (3.2) | 4 (3.5) | 6 (10.0) | 0 (0.0) | 7 (14.5) | 0.38 |
| Chronic kidney disease | 4 (6.5) | 13 (11.4) | 7 (11.6) | 6 (24.0) | 11 (22.9) | 0.09 |
| Liver disease | 2 (3.2) | 4 (3.5) | 1 (1.6) | 1 (4.0) | 1 (2.0) | 1.00 |
| Any malignancy | 0 (0.0) | 8 (7.0) | 2 (3.3) | 2 (8.0) | 4 (8.3) | 0.04 |
| Astma/COPD | 13 (21.3) | 20 (17.5) | 14 (23.3) | 5 (20.0) | 9 (18.7) | 0.72 |
| SAPS 3‡ | 53 (48, 60) | 55 (49, 60) | 57 (52, 62) | 59 (52, 63) | 56 (52, 69) | 0.18 |
| Chronic drug use | | | | | | |
| Corticosteroids§ | 5 (8.20) | 16 (14.04) | 8 (13.3) | 4 (16.0) | 6 (12.5) | 0.24 |
| Immunosuppressive therapy¶ | 1 (1.6) | 8 (7.0) | 3 (5.0) | 1 (4.0) | 1 (2.0) | 0.31 |
| Treatment limitations** | | | | | | |
| Any limitation | 14 (22.9) | 19 (16.6) | 13 (21.6) | 8 (32.0) | 13 (27.0) | 0.86 |
| No RRT | 5 (8.2) | 7 (6.1) | 5 (8.3) | 5 (20.0) | 6 (12.5) | 1.00 |
| No IMV | 6 (9.8) | 10 (8.7) | 4 (6.6) | 4 (16.0) | 6 (12.5) | 1,00 |
| No CPR | 9 (14.7) | 19 (16.6) | 12 (20.0) | 8 (32.0) | 12 (25.0) | 0.36 |
| No ECMO | 7 (11.4) | 3 (2.6) | 5 (8.3) | 3 (12.0) | 3 (6.2) | 0.15 |
| Palliative care†† | 1 (1.6) | 18 (16.5) | 7 (12.2) | 4 (16.6) | 5 (10.8) | 0.006 |

Data are n (%) or median (IQR).
*P values for the comparison between no chronic dysglycaemia and chronic dysglycaemia, Mann-Whitney U test was used for comparison of continuous data and Fisher's exact test for comparison of categorical data.
†Missing data in 15 patients (293 patients with data).
‡Missing data in 2 patients (306 patients with data).
§Systemic or inhalatory corticosteroids.
¶Immunosuppressive therapy was defined as: treatment with methotrexate, azathioprine, ciclosporin, tacrolimus, infliximab.
**Decision taken any time during ICU stay.
††Decision to go over to palliative care taken during ICU stay.
COPD, chronic obstructive pulmonary disease; CPR, cardiopulmonary resuscitation; HbA1c, glycated haemoglobin A1c; ICU, intensive care unit; IMV, invasive mechanical ventilation; OAD, oral hypoglycaemic agent; RRT, renal replacement therapy; SAPS, Simplified Acute Physiology Score.

days in the literature[16]) was sufficient to trigger new onset hyperglycaemia with mildly elevated HbA1c.

In addition to the above speculations about SARS-CoV-2 as a cause of dysglycaemia, there is also evidence suggesting that patients with pre-existing dysglycaemia are prone to a more severe course of COVID-19. For example, some studies have shown that hospitalised SARS-CoV-2 positive with pre-diabetes, unknown diabetes and known poorly controlled diabetes are at increased risk of SARS-CoV-2-associated respiratory failure requiring intensive care.[17] A higher burden of comorbidities, hyperglycaemia per se and chronic low-grade inflammation in diabetes may explain this observation.[18]

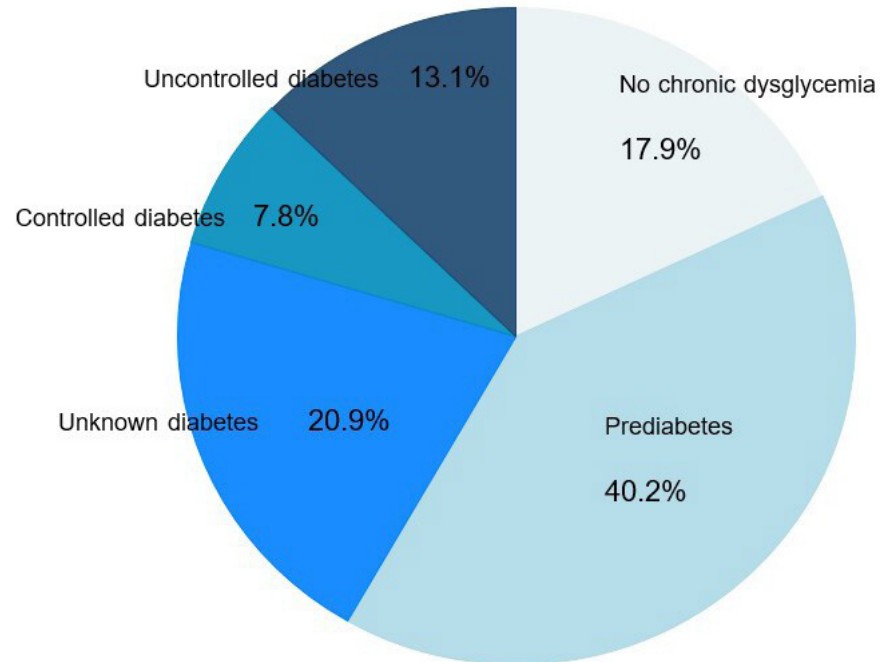

**Figure 2** Prevalence of pre-diabetes, unknown diabetes and known diabetes among 206 consecutive ICU patients with COVID-19. ICU, intensive care unit.

Wang *et al* identifiy fasting glucose as an independent predictor for 28-day mortality in hospitalised individuals with COVID-19 and previously unknown diabetes. However, HbA1c was not assessed and interference from stress hyperglycaemia might have led to the different results compared with our study.[19]

Holman *et al* identified an increased risk of death in individuals with diabetes and increasing levels of HbA1c above 48 mmol/mol and known diabetes in a large cohort of hospitalised patients, but not in critically ill individuals.[20]

Whether chronic dysglycaemia is associated with worse outcomes among COVID-19 patients admitted to ICU remains uncertain. Dennis *et al*[21] found increased mortality risk at 30 days (HR 1.23 (95% CI 1.14 to 1.32))

compared with patients with no diabetes in patients admitted to the high-dependency unit or ICU, but did not take HbA1c into consideration. A multicentre study from France including 410 ICU patients with COVID-19 found no association between the severity of dysglycaemia and tracheal intubation and/or death within 7 days of admission in patients with diabetes than in those without diabetes.[22] This is in accordance with the findings of our study. In contrast, others found higher mortality in the subgroup of mechanically ventilated patients with diabetes.[14]

We previously demonstrated an independent association between chronic dysglycaemia and need for RRT in critically ill non-COVID-19 patients.[10] This association was, however, not found in this study. In fact, we

| | | Chronic dysglycaemia | | | | |
|---|---|---|---|---|---|---|
| **Outcomes** | **No chronic dysglycaemia (n=61)** | **Pre-diabetes (n=114)** | **Unknown diabetes (n=60)** | **Controlled diabetes (n=25)** | **Uncontrolled diabetes (n=48)** | **P value\*** |
| 90-day mortality, n (%) | 9 (14.7) | 28 (24.5) | 12 (20.0) | 7 (28.0) | 15 (31.2) | 0.09 |
| ICU length of stay, days | 9 (4, 25) | 14 (6, 24) | 13 (6, 28) | 8 (5, 21) | 11 (7, 22) | 0.69 |
| Invasive mechanical ventilation, days | 16 (8, 29) | 14 (10, 23) | 15 (10, 27) | 15 (6, 21) | 14 (9, 22) | 0.60 |
| Renal replacement therapy, n (%) | 17 (27.9) | 22 (19.3) | 11 (18.3) | 1 (4.0) | 8 (16.6) | 0.06 |

**Table 2** Secondary outcomes

Data are n (%) or median (IQR).
ICU lengths of stay (4 missing values because of transfer to other hospital).
*P values for the comparison between no chronic dysglycaemia and chronic dysglycaemia, Mann-Whitney U test was used for comparison of continuous data and Fisher's exact test for comparison of categorical data.
ICU, intensive care unit.

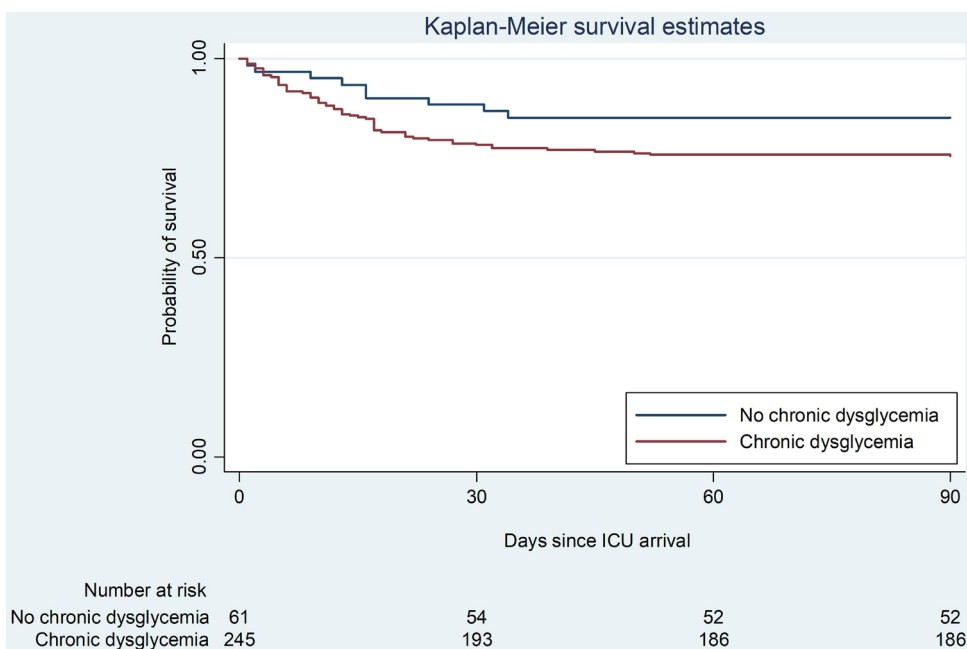

**Number at risk**

|  |  |  |  |  |
| --- | --- | --- | --- | --- |
| No chronic dysglycemia | 61 | 54 | 52 | 52 |
| Chronic dysglycemia | 245 | 193 | 186 | 186 |

**Figure 3** Probability of survival in no chronic dysglycaemia patients and in patients with chronic dysglycaemia. ICU, intensive care unit.

observed a higher proportion of patients requiring RRT among our patients without chronic dysglycaemia and an inverse association between chronic dysglycaemia and RRT use. Only one individual in the controlled diabetes subgroup received RRT during ICU stay. We believe this surprising finding may be due to treatment limitations. In fact, after exclusion of patients with treatment limitation 'not for RRT', we observed no statistically significant association between chronic dysglycaemia and RRT use. Limitations in life-sustaining care were more common in the known diabetes groups (well controlled and uncontrolled diabetes) than in all other groups. We cannot exclude the possibility that patients with severe acute or chronic kidney injury did not reach the ICU because of treatment limitation decisions made at hospital arrival or on the medical ward. This might have influenced the number of patients with kidney injury reaching the ICU, affecting predominantly patients with

**Table 3** Multivariable regression analyses showing the association of chronic dysglycaemia (vs no chronic dysglycaemia) with secondary outcomes

| Outcome measure | No chronic dysglycaemia | Chronic dysglycaemia | Adjusted risk estimate (95% CI)* | P value* | Adjusted risk estimate (95% CI)† | P value† | Statistical test |
| --- | --- | --- | --- | --- | --- | --- | --- |
| 90-day mortality n (%) | 9/61 (14.7) | 62/247 (25.1) | 1.61 (0.79 to 3.26) | 0.18 | 1.54 (0.74 to 3.19) | 0.24 | Cox regression |
| ICU length of stay, days |  |  |  |  |  |  |  |
| All patients | 9 (4, 25) | 13 (6, 23) | 1.06 (0.78 to 1.43) | 0.70 | 1.05 (0.77 to 1.44) | 0.71 | Linear regression |
| ICU survivors‡ | 9 (5, 27) | 14 (7, 24) | 1.04 (0.76 to 1.43) | 0.75 | 1.05 (0.76 to 1.44) | 0.75 | Linear regression |
| Invasive mechanical ventilation duration, days |  |  |  |  |  |  |  |
| All patients§ | 16 (8, 29) | 14 (10, 23) | 0.92 (0.68 to 1.23) | 0.58 | 0.93 (0.69 to 1.24) | 0.63 | Linear regression |
| ICU survivors¶ | 16 (8, 30) | 15 (10, 23) | 0.93 (0.70 to 1.23) | 0.61 | 0.92 (0.70 to 1.22) | 0.59 | Linear regression |
| Renal replacement therapy, n (%) |  |  |  |  |  |  |  |
| All patients | 17/61 (27.9) | 42/247 (17.0) | 0.52 (0.26 to 1.02) | 0.06 | 0.49 (0.24 to 0.99) | 0.04 | Logistic regression |
| Patients without treatment limitation as no RRT | 17/57 (29.8) | 42/224 (18.8) | 0.52 (0.26 to 1.04) | 0.10 | 0.52 (0.25 to 1.07) | 0.08 | Logistic regression |

*Multivariable models were adjusted for Simplified Acute Physiology Score (SAPS) 3, age and sex.
†Multivariable models were adjusted for SAPS 3, age, sex, hypertension, any malignancy, any treatment limitation on admission and chronic corticosteroid use.
‡ICU length of stay in ICU survivors, 260 observations.
§Invasive mechanical ventilation duration, 227 observations.
¶Invasive mechanical ventilation duration in ICU survivors, 189 observations.
ICU, intensive care unit; RRT, renal replacement therapy.

chronic dysglycaemia, as they are usually older and have multiple comorbidities.

## Strengths and limitations

Our study has several strengths. It is the first to assess the prevalence of chronic dysglycaemia in an ICU population with COVID-19 based on additional quantification of admission HbA1c. This approach reduced bias due to events that would have influenced HbA1c values obtained before ICU admission. We restricted the prevalence assessment to a cohort of patients who were admitted to ICUs where HbA1c was part of the routine laboratory panel, thereby reducing the risk of ascertainment bias. Additionally, by measuring HbA1c in all patients admitted to the ICU we identified 169 (82%) individuals with chronic dysglycaemia and 86 (41.7%) with diabetes. If HbA1c would not have been measured routinely at ICU admission, we would only have identified 43 (20.9%) individuals with diabetes. Furthermore, we considered treatment limitations in our analysis of clinical outcomes, thereby reducing the risk of treatment selection bias. Finally, we included patients admitted to ten ICUs in three University hospitals, thus providing a degree of external validity for applying our findings to similar settings.

Our study has limitations. We lack data on conditions and treatment that might have influenced admission HbA1c, such as haemoglobinopathies and blood transfusion before ICU admission. Since interviews with patients or relatives were not performed, a degree of misclassification due to non-documented dysglycaemia diagnoses cannot be ruled out. However, such interviews would have been logistically difficult during the ongoing pandemic. We used an HbA1c cut-off of 42–47 mmol/mol (6.0%–6.4%) to classify pre-diabetes. If we instead had used the cut-off suggested by the ADA (39–47 mmol/mol (5.7%–6.4%)), our prevalence of chronic dysglycaemia would have increased from 82.0% to 91.3%. This approach did not, however, alter the association with the secondary outcomes (data not shown). In addition, we lack information about glycaemic control during intensive care, which might have modified clinical outcomes.

The observational nature of the study does not imply causation. Generalisability of our results is limited to populations with similar healthcare systems and similar legal frame works for decisions on treatment limitations. Finally, the limited sample size may limit the conclusion regarding secondary outcomes that can be drawn from the data.

## CONCLUSION

In our multicentre cohort of COVID-19 patients admitted to the ICU, HbA1c screening diagnosed chronic dysglycaemia in four out of five patients with the majority having either pre-diabetes or previously undiagnosed diabetes. Chronic dysglycaemia was not significantly associated with mortality, ICU length of stay, duration of IMV or RRT use after considering treatment limitations. These findings indicate that chronic dysglycaemia may be a risk factor for severe COVID-19. However, COVID-19 prognosis in the ICU does not appear to be modified by chronic dysglycaemia.

**Contributors** AB, SR, MC, JG, AO, CS and JM contributed to the concept and design of the study. AB, SR, CH and JM collected data. AB, SR, CH, JG, MC, CS, AO and JM contributed to the analysis and interpretation of data. AB and JM drafted the manuscript. All authors critical reviewed and approved the final manuscript. AB accepts full responsibility for the work and the conduct of the study, had access to the data, and controlled the decision to publish. AB is the guarantor for the work and conduct of the study.

**Funding** AB and JM were supported by Region Stockholm (clinical research appointment and ALF project grants), grant number 580282.

**Competing interests** None declared.

**Patient and public involvement** Patients and/or the public were not involved in the design, or conduct, or reporting, or dissemination plans of this research.

**Patient consent for publication** Not applicable.

**Ethics approval** The study was approved by the Swedish Ethical Review Authority (approval number 2020-01302, amendment 2020-02890) with a waiver of informed consent.

**Provenance and peer review** Not commissioned; externally peer reviewed.

**Data availability statement** Data are available on reasonable request. The data that support the findings of this study are available from the corresponding author on reasonable request.

**ORCID iD**
Anca Balintescu http://orcid.org/0000-0001-8665-5742

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
