## [Reviewer comments · BMJ Open]

ARTICLE DETAILS

TITLE (PROVISIONAL)	Prevalence and impact of chronic dysglycemia among patients with Covid-19 in Swedish intensive care units: a multicenter, retrospective cohort study
AUTHORS	Balintescu, Anca; Rysz, Susanne; Hertz, Carl; Grip, Jonathan; Cronhjort, Maria; Oldner, Anders; Svensen, Christer; Mårtensson, Johan

VERSION 1 – REVIEW

REVIEWER	Mehta, Yatin Anaesthesiology and Critical Care, Medanta – The Medicity
REVIEW RETURNED	24-Jan-2023

GENERAL COMMENTS	Comments 1: Title mentions this as a “retrospective cohort study”, whereas at Page 3, line 17, it is mentioned that study design is “Multicenter prospective observational study”. This needs clarification. Comments 2: In the statistical analysis section, following test are mentioned as used for analysis: • Chi-square test or Fisher's exact test for comparing percentages• Mann-Whitney U test (for two groups) or the Kruskal Wallis test (for multiple groups). In the results, presented in Tables 1 and 2 it appears Chi square test have been used for comparing proportions and Kruskal Wallis test for numerical data (for multiple groups). Where Fisher's exact test and Mann-Whitney U test (for two groups) have been used, should be spelled out. Comments 3: For important outcome parameters such as 90 day mortality, the p – value is given comparing all groups. It is observed that 90 day mortality is highest (31.2%) for uncontrolled diabetes as compared to no chronic dysglycaemia (14.7%). Similarly, renal replacement therapy, the proportion is 27.9% for no chronic dysglycaemia and just 4.0% for controlled diabetes. In such situations, the pair wise comparisons should have been attempted. Comments 4: In the analysis using Multivariable regression analyses following is mentioned: “We observed a trend towards higher mortality (adjusted HR 1.61, 95% CI 0.79-3.26, P=0.18) and lower RRT use (adjusted OR 0.52, 95% CI 0.26-1.02, P=0.06) in patients with chronic dysglycemia (Table 3)”. However the results presented are only for two groups
--

	namely “No chronic dysglycaemia and chronic dysglycaemia”. The trend could not be seen with this analysis. In the Multivariable regression analyses, it is mentioned that the results were adjusted for Simplified Acute Physiology Score (SAPS) 3, age and sex. Ideally the cofactors such as hypertension, any malignancy and palliative care should also have been considered which were observed significant in Table 2. Comments 5: The findings based on Figure S5. Probability of survival in the study groups should have been discussed in the result section. Comments 6: In the section on Strengths and limitations of this study, the specifics mentioned are not clear. It should be clearly spelt out.
--	---

REVIEWER	van Baall, Lukas University of Duisburg-Essen
REVIEW RETURNED	07-Feb-2023

GENERAL COMMENTS	The study by Balintescu et al., titled 'I' aimed to document the prevalence of chronic dysglycemia in critically ill patients with Covid-19, as well as the association between Covid-19 related outcomes and glycemic status. They studied critically ill hospitalized patients with COVID-19 on multiple ICU-wards with systematical screening at admission for diabetes and prediabetes by HbA1c and known history of diabetes. After presenting the results, they conclude that the prevalence of dysglycemia in critically ill Covid-19 patients is higher than described in previous studies. Moreover, they emphasize that there is no significant association between present dysglycemia and Covid-19 related outcomes. This is an interesting manuscript because it shows that the prevalence of diabetes is higher in critically ill COVID19 patients than generally believed. However, I have some concerns that make publication difficult in its current state. 1. Definition of Prediabetes, defined in this study by a HbA1c-value of 42-47 mmol/mol [6.0-6.4%], is not comprehensible. Generally prediabetes is defined by a HbA1c-value of 5.7%–6.4% (39–47 mmol/mol) [Data from American Diabetes Association. 2. Classification and diagnosis of diabetes: standards of medical care in diabetes-2019. Diabetes Care. 2019;42:S13–s28]. Even in the cited WHO-classification prediabetes is not defined as described in the study. In consequence patients with prediabetes are misclassified as having no chronic dysglycemia and therefore may hamper the results. I strongly recommend to perform analysis using a prediabetes definition as recommended by the ADA 2. The cut-off value for unknown diabetes does not correspond. unknown diabetes Do the authors mean a HbA1c of ≥ 48 mmol/mol [6.5 %] or ≥ 52 mmol/mol [6.9 %]? Please clarify. 3. Confounding factors that may influence disease outcome (such as glucocorticoid, remdesivir, statins, antidiabetics, antihypertensives) were not provided and should be taken in account for the regression analysis, as well as BMI 4. Did the authors screened medical records for the diagnosis of diabetes or also for antidiabetic information. Did they perform a personal interview with the patients after
--

	ICU-discharge to exclude bias of non documented dysglycemia diagnosis? 5. The authors provide subpopulations in the group of patients with chronic dysglycemia. I would recommend to perform subgroup analyses between patients without chronic dysglycemia and patients with diabetes/controlled diabetes and uncontrolled diabetes 6. After statistical assuring, the association between dysglycemia and disease outcomes may be discussed in more detail, especially as numerous studies already demonstrated an adverse association between Covid-19 related outcomes and dysglycemia (Holman N et al. PMID: 32798471 / Montefusco et al. PMID: 34035524 / Apicella et al. PMID: 32687793 / Wang et al. PMID: 32647915 / Dennis et al. PMID: 33097559) 7. As provided by the authors, there is a lack of information about the glycemic control during the ICU-stay. The ADA defines diabetes in a patient with classic symptoms of hyperglycemia or hyperglycemic crisis, by a random PG ≥ 200 mg/dL (11.1 mmol/L). Did any of the patients receive a measurement of plasma glucose at admission? If yes, the values should be taken in account for defining the patients glycemic status and the calculations should be adjusted. 8. An unique selling point of this study is the systematic measurement of HbA1c. Nevertheless, it should be discussed in more detail, how the systematic use of HbA1c-measurement influences the prevalence of dysglycemia
--	--

REVIEWER	Laviada-Molina, Hugo A.
REVIEW RETURNED	17-Feb-2023

GENERAL COMMENTS	I think this work has the merits for being published. I would like more extensive comments about the limitations of the design and the sample, to address the research question.
--

VERSION 1 – AUTHOR RESPONSE

Reviewer: 1

Dr. Yatin Mehta, Anaesthesiology and Critical Care, Medanta – The Medicity

Comments to the Author:

To be reviewed after addressing the issues in the attached file.

*Please see the attached report from this reviewer

BMJopne2022-071330

Comments 1:

Title mentions this as a “retrospective cohort study”, whereas at Page 3, line 17, it is mentioned that study design is “Multicenter prospective observational study”. This needs clarification.

Response: We agree with reviewer #1 that this is confusing and needs clarification. The study follows patients from admission to the ICU prospectively until 90 days after ICU admission or death. However, data was gathered and analyses were done retrospectively. We have now changed the text in the revised Abstract and Method sections that now states “multicenter retrospective observational study” (Page 2 and page 5, last paragraph)

Comments 2:

In the statistical analysis section, following test are mentioned as used for analysis:

- Chi-square test or Fisher's exact test for comparing percentages
- Mann-Whitney U test (for two groups) or the Kruskal Wallis test (for multiple groups).

In the results, presented in Tables 1 and 2 it appears Chi square test have been used for comparing proportions and Kruskal Wallis test for numerical data (for multiple groups). Where Fisher's exact test and Mann-Whitney U test (for two groups) have been used, should be spelt out.

Response: We thank reviewer 1# for this comment. We have now decided to use the Fisher’s exact test for comparing categorical data, and the Mann-Whitney U test to compare the “no chronic dysglycemia” vs “chronic dysglycemia” groups for continuous data, throughout the manuscript. The p values are presented in Table 1 and 2. This is clarified in the footnotes to the revised Table 1 and Table 2 (page 9 and 11) and in the revised Method section (page 7, last paragraph)

Table 1. Baseline characteristics and treatment limitations

Characteristic	No chronic dysglycemia	Chronic dysglycemia				P ^e
		Prediabetes	Unknown diabetes	Controlled diabetes	Uncontrolled diabetes	
No. (%)	61 (19.8)	114 (37.0)	60 (19.4)	25 (8.11)	48 (15.5)	
Age, years	57 (51,63)	61 (53, 68)	60 (52, 68)	63 (57, 71)	62 (55,69)	0.03
Male sex	48 (78.6)	92 (80.7)	47 (78.3)	21 (84.0)	36 (75.0)	1.00
Body mass index ^a , kg/m ²	27 (25, 32)	27 (25, 30)	28 (25, 31)	29 (26, 32)	30 (26, 33)	0.97
HbA1c, mmol/mol	39 (36, 40)	44 (43, 46)	51 (49, 57)	47 (44, 49)	70 (61, 81)	<0.001

Diabetes treatment						
Diet only				6 (24.0)	2 (4.1)	
OAD only				17 (68)	19 (39.5)	
Insulin only				1 (4.0)	12 (25.0)	
OAD+Insulin				1 (4.0)	15 (31.2)	
Comorbidity						
Hypertension	18 (29.5)	40 (35.0)	23 (38.3)	16 (64.0)	34 (70.8)	0.02
Heart failure	6 (9.8)	5 (4.3)	6 (10.0)	0 (0.0)	3 (6.2)	0.24
Previous myocardial infarction	2 (3.2)	4 (3.5)	6 (10.0)	0 (0.0)	7 (14.5)	0.38
Chronic kidney disease	4 (6.5)	13 (11.4)	7 (11.6)	6 (24.0)	11 (22.9)	0.09
Liver disease	2 (3.2)	4 (3.5)	1 (1.6)	1 (4.0)	1 (2.0)	1.00
Any malignancy	0 (0.0)	8 (7.0)	2 (3.3)	2 (8.0)	4 (8.3)	0.04
Astma/COPD	13 (21.3)	20 (17.5)	14 (23.3)	5 (20.0)	9 (18.7)	0.72
SAPS 3 ^b	53 (48, 60)	55 (49, 60)	57 (52, 62)	59 (52, 63)	56 (52, 69)	0.18
Chronic drug use						
Corticosteroids ^c	5 (8.20)	16 (14.04)	8 (13.3)	4 (16.0)	6 (12.5)	0.24
Immunosuppressive therapy ^d	1 (1.6)	8 (7.0)	3 (5.0)	1 (4.0)	1(2.0)	0.31
Treatment limitations ^f						
Any limitation	14 (22.9)	19 (16.6)	13 (21.6)	8 (32.0)	13 (27.0)	0.86
No RRT	5 (8.2)	7 (6.1)	5 (8.3)	5 (20.0)	6 (12.5)	1.00
No IMV	6 (9.8)	10 (8.7)	4 (6.6)	4 (16.0)	6 (12.5)	1,00
No CPR	9 (14.7)	19 (16.6)	12 (20.0)	8 (32.0)	12 (25.0)	0.36
No ECMO	7 (11.4)	3 (2.6)	5 (8.3)	3 (12.0)	3 (6.2)	0.15
Palliative care ^g	1 (1.6)	18 (16.5)	7 (12.2)	4 (16.6)	5 (10.8)	0.006

Data are n (%) or median (interquartile range).

HbA1c, glycated hemoglobin A1c; OAD, oral hypoglycemic agent; COPD, Chronic obstructive pulmonary disease; SAPS, Simplified Acute Physiology Score, 2 missing values (306 patients with

data); RRT, renal replacement therapy; IMV, invasive mechanical ventilation; CPR, cardiopulmonary resuscitation

^aMissing data in 15 patients (293 patients with data)

^bMissing data in 2 patients (306 patients with data)

^cSystemic or inhaled corticosteroids

^dImmunosuppressive therapy was defined as: treatment with Metotrexate, Azathioprin, Ciklosporin, Tacrolimus, Infliximab

^eP values for the comparison between no chronic dysglycemia and chronic dysglycemia, Mann-Whitney U test was used for comparison of continuous data and Fischer's exact test for comparison of categorical data

^fDecision taken any time during ICU stay

^gDecision to go over to palliative care taken during ICU stay

Table 2. Secondary outcomes

Outcomes	No chronic dysglycemia (n = 61)	Chronic dysglycemia				P ^a
		Prediabetes (n = 114)	Unknown diabetes (n = 60)	Controlled diabetes (n = 25)	Uncontrolled diabetes (n = 48)	
90-day mortality, n (%)	9 (14.7)	28 (24.5)	12 (20.0)	7 (28.0)	15 (31.2)	0.09
ICU length of stay, days	9 (4, 25)	14 (6, 24)	13 (6, 28)	8 (5, 21)	11 (7, 22)	0.69
Invasive mechanical ventilation, days	16 (8, 29)	14 (10, 23)	15 (10, 27)	15 (6, 21)	14 (9, 22)	0.60
Renal replacement therapy, n (%)	17 (27.9)	22 (19.3)	11 (18.3)	1 (4.0)	8 (16.6)	0.06

Data are n (%) or median (interquartile range)

ICU lengths of stay (4 missing values because of transfer to other hospital)

^a P values for the comparison between no chronic dysglycemia and chronic dysglycemia, Mann-Whitney U test was used for comparison of continuous data and Fischer's exact test for comparison of categorical data.

Comments 3:

For important outcome parameters such as 90 day mortality, the p – value is given comparing all groups. It is observed that 90 day mortality is highest (31.2%) for uncontrolled diabetes as compared to no chronic dysglycaemia (14.7%). Similarly, renal replacement therapy, the

proportion is 27.9% for no chronic dysglycaemia and just 4.0% for controlled diabetes. In such situations, the pair wise comparisons should have been attempted.

Response: We agree with reviewer #1 that a pairwise comparison between groups would be appropriate. We therefore performed a post-hoc pairwise comparison for 90 day mortality and renal replacement therapy between the study subgroups. This is presented in Table S1 in the revised Supplementary Appendix and referred to in the revised Method (page 8, first paragraph) and Result (page 13, second paragraph) sections.

Table S1: Post-hoc exploratory comparison between the subgroups for 90 day mortality and renal replacement therapy

Subgroups	90 day mortality		Renal replacement therapy	
	N (%)	p	N (%)	p
No chronic dysglycemia vs Prediabetes	9/61 (14.7) vs 28/114 (24.5)	0.17	17/61 (27.8) vs 22/114 (19.2)	0.25
No chronic dysglycemia vs Unknown diabetes	9/61 (14.7) vs 12/60 (20.0)	0.48	17/61 (27.8) vs 11/60 (18.3)	0.28
No chronic dysglycemia vs Controlled diabetes	9/61 (14.7) vs 7/25 (28.0)	0.22	17/61 (27.8) vs 1/25 (4.0)	0.01
No chronic dysglycemia vs Uncontrolled diabetes	9/61 (14.7) vs 15/48 (31.2)	0.06	17/61 (27.8) vs 8/48 (16.6)	0.25
Prediabetes vs Unknown diabetes	28/114 (24.5) vs 12/60 (20.0)	0.57	22/114 (19.2) vs 11/60 (18.3)	1
Prediabetes vs Controlled diabetes	28/114 (24.5) vs 7/25 (28.0)	0.79	22/114 (19.2) vs 1/25 (4.0)	0.07
Prediabetes vs Uncontrolled diabetes	28/114 (24.5) vs 15/48 (31.2)	0.43	22/114 (19.2) vs 8/48 (16.6)	0.8
Unknown diabetes vs Controlled diabetes	12/60 (20.0) vs 7/25 (28.0)	0.41	11/60 (18.3) vs 1/25 (4.0)	0.1
Unknown diabetes vs Uncontrolled diabetes	12/60 (20.0) vs 15/48 (31.2)	0.18	11/60 (18.3) vs 8/48 (16.6)	1

Controlled vs Uncontrolled diabetes	7/25 (28.0) vs 15/48 (31.2)	0.77	1/25 (4.0) vs 8/48 (16.6)	0.01
No chronic dysglycemia and prediabetes vs unknown and known diabetes	37/175 (21.1) vs 34/133 (25.5)	0.41	39/175 (22.2) vs 20/133 (15.0)	0.14

P values calculated with Fischer's exact test

RRT use was higher in the no diabetes group compared to the controlled diabetes group as well as in the uncontrolled diabetes compared to controlled diabetes group (Table S1) in the post-hoc exploratory comparison between the subgroups.

Comments 4:

In the analysis using Multivariable regression analyses following is mentioned:

"We observed a trend towards higher mortality (adjusted HR 1.61, 95% CI 0.79-3.26, P=0.18) and lower RRT use (adjusted OR 0.52, 95% CI 0.26-1.02, P=0.06) in patients with chronic dysglycemia (Table 3)". However the results presented are only for two groups namely "No chronic dysglycaemia and chronic dysglycaemia". The trend could not be seen with this analysis.

In the Multivariable regression analyses, it is mentioned that the results were adjusted for Simplified Acute Physiology Score (SAPS) 3, age and sex. Ideally the cofactors such as hypertension, any malignancy and palliative care should also have been considered which were observed significant in Table 2.

Response: We agree with reviewer 1# that the word "trend" is not adequate. We have now replaced the text in the revised Result section, that states: "*On multivariable regression analysis we observed a numerically higher mortality (adjusted HR 1.54, 95% CI 0.74-3.19, P=0.24) and significantly lower RRT use (adjusted OR 0.49, 95% CI 0.24-0.99, P=0.04) in patients with chronic dysglycemia (Table 3).*" (page 12, last paragraph)

In response to the questions raised by reviewer #1 and reviewer #2, we have added the following analysis: In the multivariable regression analysis we have now additionally adjusted for hypertension, any malignancy, any treatment limitation on admission and chronic corticosteroid use. In our cohort, 71 patients died within 90 days from ICU admission. Following the thumb rule of adjusting for one covariate for every 10 observations of outcome, we could adjust for 7 covariates in total. After careful consideration we decided to adjust for: SAPS 3, age, sex, hypertension, any malignancy, chronic corticosteroids and any treatment limitation on admission. Individuals with treatment limitations set during ICU stay might have been preceded by different periods of full medical care and decision to step down treatment is usually taken when patients are terminally ill. We therefore decided to adjust for any treatment limitation set at ICU admission instead.

Results are presented in the revised Table 3, where we present risk estimates for a model where we adjusted for SAPS 3, age, sex and another model where we adjusted for all 7 covariates.

Table 3. Multivariable regression analyses showing the association of chronic dysglycemia (versus no chronic dysglycemia) with secondary outcomes

Outcome measure	No Chronic Dysglycemia	Chronic Dysglycemia	Adjusted Risk Estimate (95% CI) ^a	P ^a	Adjusted Risk Estimate (95% CI) ^b	P ^b	Statistical test
90-day mortality n (%)	9/61 (14.7)	62/247 (25.1)	1.61 (0.79 to 3.26)	0.18	1.54 (0.74 to 3.19)	0.24	Cox regression
ICU length of stay, days							
All patients	9 (4, 25)	13 (6, 23)	1.06 (0.78 to 1.43)	0.70	1.05 (0.77 to 1.44)	0.71	Linear regression
ICU survivors ^c	9 (5, 27)	14 (7, 24)	1.04 (0.76 to 1.43)	0.75	1.05 (0.76 to 1.44)	0.75	Linear regression
Invasive mechanical ventilation duration, days							
All patients ^d	16 (8, 29)	14 (10, 23)	0.92 (0.68 to 1.23)	0.58	0.93 (0.69 to 1.24)	0.63	Linear regression
ICU survivors ^e	16 (8, 30)	15 (10, 23)	0.93 (0.70 to 1.23)	0.61	0.92 (0.70 to 1.22)	0.59	Linear regression
Renal replacement therapy, n (%)							

All patients	17/61 (27.9)	42/247 (17.0)	0.52 (0.26 to 1.02)	0.06	0.49 (0.24 to 0.99)	0.04	Logistic regression
Patients without treatment limitation as no RRT	17/57 (29.8)	42/224 (18.8)	0.52 (0.26 to 1.04)	0.10	0.52 (0.25 to 1.07)	0.08	Logistic regression

^aMultivariable models were adjusted for Simplified Acute Physiology Score (SAPS) 3, age and sex.

^bMultivariable models were adjusted for Simplified Acute Physiology Score (SAPS) 3, age, sex, hypertension, any malignancy, any treatment limitation on admission and chronic corticosteroid use

^cICU length of stay in ICU survivors, 260 observations

^dInvasive mechanical ventilation duration, 227 observations

^eInvasive mechanical ventilation duration in ICU survivors, 189 observations

When adjusting for the 7 possible confounders, chronic dysglycemia was associated with lower use of RRT (HR 0.49, 95 % CI 0.24 to 0.99, P=0.04), however this association was not significant when patients with treatment limitations were excluded (HR 0.52, 95% CI 0.25 to 1.07, P=0.08).

Comments 5:

The findings based on Figure S5. Probability of survival in the study groups should have been discussed in the result section.

Response: We agree with reviewer 1# that further explanations is required. We therefore have mentioned these results in the revised Result section: *“Individuals with uncontrolled diabetes had the lowest probability of survival followed by individuals with controlled diabetes and prediabetes. The highest probability of survival was observed among patients with no chronic dysglycemia and prediabetes, respectively (Figure S5). However, we observed no statistically significant differences in mortality in the post-hoc comparison of subgroups (Table S1).”* (page 13, second paragraph).

Comments 6:

In the section on Strengths and limitations of this study, the specifics mentioned are not clear. It should be clearly spelt out.

Response: We agree with reviewer 1# and we have now extended the strengths and limitation section in the revised Discussion section that states: *“Our study has limitations. We lack data on conditions and treatment that might have influenced admission HbA1c, such as haemoglobinopathies and blood transfusion before ICU admission. Since interviews with patients or relatives were not performed, a degree of misclassification due to non-documented dysglycemia diagnoses cannot be ruled out. However, such interviews would have been logistically difficult during the ongoing pandemic. We used an HbA1c cutoff of 42-47 mmol/mol (6.0-6.4%) to classify prediabetes. If we instead had used the*

cutoff suggested by the American Diabetes Association (39-47 mmol/mol [5.7-6.4%], our prevalence of chronic dysglycemia would have increased from 82.0% to 91.3%. This approach did not, however, alter the association with the secondary outcomes (data not shown). In addition, we lack information about glycemic control during intensive care, which might have modified clinical outcomes.

The observational nature of the study does not imply causation. Generalizability of our results is limited to populations with similar health care systems and similar legal frame-works for decisions on treatment limitations. Finally, the limited sample size may limit the conclusion regarding secondary outcomes that can be drawn from the data.” (page 17, second paragraph).

Reviewer: 2

Dr. Lukas van Baall, University of Duisburg-Essen

Comments to the Author:

The study by Balintescu et al., titled 'I' aimed to document the prevalence of chronic dysglycemia in critically ill patients with Covid-19, as well as the association between Covid-19 related outcomes and glycemic status. They studied critically ill hospitalized patients with COVID-19 on multiple ICU-wards with systematical screening at admission for diabetes and prediabetes by HbA1c and known history of diabetes. After presenting the results, they conclude that the prevalence of dysglycemia in critically ill Covid-19 patients is higher than described in previous studies. Moreover, they emphasize that there is no significant association between present dysglycemia and Covid-19 related outcomes.

This is an interesting manuscript because it shows that the prevalence of diabetes is higher in critically ill COVID19 patients than generally believed. However, I have some concerns that make publication difficult in its current state.

1. Definition of Prediabetes, defined in this study by a HbA1c-value of 42-47 mmol/mol [6.0-6.4%], is not comprehensible. Generally prediabetes is defined by a HbA1c-value of 5.7%–6.4% (39–47 mmol/mol) [Data from American Diabetes Association. 2. Classification and diagnosis of diabetes: standards of medical care in diabetes-2019. Diabetes Care. 2019;42:S13–s28]. Even in the cited WHO-classification prediabetes is not defined as described in the study. In consequence patients with prediabetes are misclassified as having no chronic dysglycemia and therefore may hamper the results. I strongly recommend to perform analysis using a prediabetes definition as recommended by the ADA

Response: We agree with reviewer #2 that the definition of prediabetes according to WHO is not clearly stipulated. We considered the cut off values endorsed by WHO (according to reference 12, page 8, third paragraph): “While recognizing the continuum of risk that may be captured by the HbA1c assay, the International Expert Committee recommended that persons with a HbA1c level between 6.0 and 6.5% were at particularly high risk and might be considered for diabetes prevention interventions.” Additionally, these are the cut off values for the diagnosis of prediabetes in Sweden and permits comparison with a similar study in a general ICU population before the Covid-19 pandemic (Balintescu et al. Prevalence and impact of chronic dysglycemia in intensive care unit patients—A retrospective cohort study. *Acta Anaesthesiol Scand.* 2020; 65: 82– 91. <https://doi.org/10.1111/aas.13695>).

However, there were 31 patients classified as “no diabetes” that would have been included in the prediabetes group when using the ADA classification. Prevalence of chronic dysglycemia would have been 91.3% (188 individuals) with 49.5% (102 individuals) having prediabetes. No significant changes in secondary outcomes were observed when analysing the data using the ADA classification.

Accordingly, we have added the following statement to the limitations section (page 17, second paragraph): “*We used an HbA1c cutoff of 42-47 mmol/mol (6.0-6.4%) to classify prediabetes. However, if we had used the cutoff suggested by the American Diabetes Association (39-47 mmol/mol [5.7-6.4%], our prevalence of chronic dysglycemia would have increased from 82.0% to*

91.3%. *This approach did not, however, alter the association with the secondary outcomes (data not shown).*”

2. The cut-off value for unknown diabetes does not correspond. unknown diabetes
Do the authors mean a HbA1c of ≥ 48 mmol/mol [6.5 %] or ≥ 52 mmol/mol [6.9 %]?
Please clarify.

Response: We thank reviewer #2 for noticing this error. We have now corrected the cut-off values for HbA1c for the subgroup with unknown diabetes. We defined unknown diabetes as a HbA1c value equal to or above 48 mmol/mol [6.5 %] at admission without previous diabetes history. This is now specified under the revised Method section (page 7, first paragraph).

3. Confounding factors that may influence disease outcome (such as glucocorticoid, remdesivir, statins, antidiabetics, antihypertensives) were not provided and should be taking in account for the regression analysis, as well as BMI

Response: We agree that some additional covariates should have been adjusted for. Due to the number of outcomes in the mortality analysis, we could adjust for 7 covariates. Based on the concern raised by reviewer #1 and reviewer #2, we have therefore included an additional model adjusting for the following covariates: SAPS 3, age, sex, hypertension, any malignancy, corticosteroids and treatment limitations at ICU admission. None of patients received remdesivir as this treatment option was not available during the first wave of Covid-19 in Sweden. We had no data regarding statins or antihypertensive medications. We did not adjust for oral antidiabetic medications or insulin as only individuals with controlled and uncontrolled diabetes could receive these treatments. Due to the limited number of covariates we could adjust for, missing data for BMI, and due to the non-significant difference between the chronic dysglycemia and no chronic dysglycemia groups for BMI in Table 1, we decided not to adjust for BMI in the analyses.

4. Did the authors screened medical records for the diagnosis of diabetes or also for antidiabetic information. Did they perform a personal interview with the patients after ICU-discharge to exclude bias of non documented dysglycemia diagnosis?

Response: The patient's medical records were screened for diabetes diagnosis registered prior to ICU admission. Screening was done by opening each journal and searching for ICD codes for diabetes E08-E13 (Diabetes mellitus - ICD-10 Codes- Codify by AAPC). Additionally, the patient's medical record was screened for oral antidiabetic treatment, GLP1-RAs, and insulin treatment prior to ICU admission.

Interviews with patients or relatives were not performed and a degree of misclassification due to non-documented dysglycemia diagnoses cannot be ruled out. However, such interviews would have been logistically difficult during the ongoing pandemic. This limitation is now mentioned in the revised Discussion section (page 17, second paragraph).

5. The authors provide subpopulations in the group of patients with chronic dysglycemia. I would recommend to perform subgroup analyses between patients without chronic dysglycemia and patients with diabetes/controlled diabetes and uncontrolled diabetes

Response: We have now addressed this issued in the post-hoc exploratory pairwise comparison analysis (Table S1).

Table S1: Post-hoc exploratory comparison between the subgroups for 90 days mortality and Renal replacement therapy

Subgroups	90 day mortality		Renal replacement therapy	
	N (%)	p	N (%)	p
No chronic dysglycemia vs Prediabetes	9/61 (14.7) vs 28/114 (24.5)	0.17	17/61 (27.8) vs 22/114 (19.2)	0.25
No chronic dysglycemia vs Unknown diabetes	9/61 (14.7) vs 12/60 (20.0)	0.48	17/61 (27.8) vs 11/60 (18.3)	0.28
No chronic dysglycemia vs Controlled diabetes	9/61 (14.7) vs 7/25 (28.0)	0.22	17/61 (27.8) vs 1/25 (4.0)	0.01
No chronic dysglycemia vs Uncontrolled diabetes	9/61 (14.7) vs 15/48 (31.2)	0.06	17/61(27.8) vs 8/48 (16.6)	0.25
Prediabetes vs Unknown diabetes	28/114 (24.5) vs 12/60 (20.0)	0.57	22/114 (19.2) vs 11/60 (18.3)	1
Prediabetes vs Controlled diabetes	28/114 (24.5) vs 7/25 (28.0)	0.79	22/114 (19.2) vs 1/ 25 (4.0)	0.07
Prediabetes vs Uncontrolled diabetes	28/114 (24.5) vs 15/48 (31.2)	0.43	22/114 (19.2) vs 8/48 (16.6)	0.8
Unknown diabetes vs Controlled diabetes	12/60 (20.0) vs 7/25 (28.0)	0.41	11/60 (18.3) vs 1/25 (4.0)	0.1
Unknown diabetes vs Uncontrolled diabetes	12/60 (20.0) vs 15/48 (31.2)	0.18	11/60 (18.3) vs 8/48 (16.6)	1
Controlled vs Uncontrolled diabetes	7/25 (28.0) vs 15/48 (31.2)	0.77	1/25 (4.0) vs 8/48 (16.6)	0.01
No chronic dysglycemia and prediabetes vs unknown and known diabetes	37/175 (21.1) vs 34/133 (25.5)	0.41	39/175 (22.2) vs 20/133 (15.0)	0.14

P values calculated with Fischer's exact test

6. After statistical assuring, the association between dysglycemia and disease outcomes may be discussed in more detail, especially as numerous studies already demonstrated an adverse association between Covid-19 related

outcomes and dysglycemia (Holman N et al. PMID: 32798471 / Montefusco et al. 34035524 / Apicella et al. PMID: 32687793 / Wang et al. PMID: 32647915 / Dennis et al. PMID: 33097559)

Response: We agree that this could be discussed in more detail. Our results refer exclusively to individuals with Covid-19 admitted to the ICU. Additionally, we also assessed patients with prediabetes previous to ICU admission. Holman et al investigated individuals with established diabetes diagnosis in the general population and found an association between death and increasing levels of HbA1c above 48 mmol/mol. Additionally, HbA1c was measured in a time frame of 1 year prior to Covid-19 diagnosis. In our study 90-day mortality was numerically higher in the subgroup with uncontrolled diabetes compared to the controlled diabetes and no chronic dysglycemia groups, but no statistically significant association of chronic dysglycemia with mortality could be observed. We cannot exclude that the small sample size precluded such an observation.

Wang et al identifies fasting glucose as an independent predictor for 28-day mortality in hospitalised individuals with Covid-19 and no previously known diabetes. However, HbA1c was not assessed and interference from stress hyperglycemia might have led to the different results compared to our study.

Montefusco et al, showed increased mortality in individuals with known and newly diagnosed diabetes compared to those without glycemic perturbances (HR: 2.16 CI: 1.27–3.67, p=0.009) or in those with new-onset hyperglycemia (HR: 2.05 CI: 1.28–3.29, p=0.002), in a cohort of hospitalized patients in Italy. Overall, 27% had diabetes according to HbA1c criteria, with more than one third of patients in this group (65 out of 151) had newly diagnosed diabetes. These were however hospitalized patients and no information is given about the ICU population. This might explain the lower prevalence of diabetes in this cohort. Additionally, prediabetes according to HbA1c criteria was not considered.

Another cohort of patients admitted to the High Dependency Unit and ICU with previously known type 2 diabetes and Covid-19 found increased mortality risk at 30 days (HR 1.23 [95% CI 1.14, 1.32]) compared to patients with no diabetes. However, no measure of chronic glycemic control was reported and the study evaluated only individuals with known diabetes. Additionally, 30 day mortality was much higher in both the ICU group and no diabetes group (30-day mortality 45.2% [42.4, 47.9] and 36.3% [34.7, 37.8]) compared to our study. Higher mortality rates might indicate that treatment limitations were not considered.

We have now extended the Discussion section and commented on some of the mentioned studies (page 15, second paragraph).

7. As provided by the authors, there is a lack of information about the glycemic control during the ICU-stay. The ADA defines diabetes in a patient with classic symptoms of hyperglycemia or hyperglycemic crisis, by a random PG ≥ 200 mg/dL (11.1 mmol/L). Did any of the patients receive a measurement of plasma glucose at admission? If yes, the values should be taken in account for defining the patients glycemic status and the calculations should be adjusted.

Response: We agree with reviewer #2 that a single, random plasma glucose level ≥ 200 mg/dL (11.1 mmol/L) with concomitant symptoms for hyperglycemia or hyperglycemic crisis can establish diabetes diagnosis according to ADA in non critically ill individuals. However, patients referred to the ICU are critically ill and have different degrees of stress hyperglycemia unrelated to a chronic alteration in glycemic metabolism. Using only plasma glucose levels for the diagnosis of diabetes in critically ill might introduce bias, classifying individuals with stress hyperglycemia as suffering of diabetes when this is not the case. Additionally, HbA1c is not altered by the onset of critical illness (Luethi et al, Glycated Hemoglobin A1c Levels Are Not Affected by Critical Illness. Crit Care Med. 2016 Sep;44(9):1692-4). The aim of our study was to determine the prevalence of individuals with

chronically altered glucose metabolism and Covid-19 that are admitted to the ICU. We agree that admission plasma glycemia levels would have contributed to identify individuals with stress hyperglycemia in the context of Covid-19 and critical illness, but this was not the main focus of the paper.

8. An unique selling point of this study is the sytemical measurement of HbA1c.
 Nevertheless, it should be discussed in more detail, how the systematic use of HbA1c-measurment influences the prevalence of dysglycemia.

Response: We agree that this is important. We have now extended the explanation regarding the systemic HbA1c measurement. The revised Discussion section now states: *“We restricted the prevalence assessment to a cohort of patients who were admitted to ICUs where HbA1c was part of the routine laboratory panel, thereby reducing the risk of ascertainment bias. Additionally, by measuring HbA1c in all patients admitted to the ICU we identified 169 (82%) individuals with chronic dysglycemia and 86 (41.7%) had diabetes. If HbA1c would not have been measured routinely at ICU admission, we would only have identified 43 (20.9%) individuals with diabetes.”* (page15, last paragraph)

Reviewer: 3
 Dr. Hugo A. Laviada-Molina
 Comments to the Author:

I think this work has the merits for being published. I would like more extensive comments about the limitations of the design and the sample, to adress the research question.

Response: We agree with reviewer #3 that limitations due to sample size and design should be mentioned. We have now extended the limitations section in the revised Discussion section that now states: *“The observational nature of the study does not imply causation. Generalizability of our results is limited to populations with similar health care systems and similar legal frame-works for decisions on treatment limitations. Finally, the limited sample size may limit the conclusion regarding secondary outcomes that can be drawn from the data.”* (page 17, last paragraph)

VERSION 2 – REVIEW

REVIEWER	Mehta, Yatin Anaesthesiology and Critical Care, Medanta – The Medicity
REVIEW RETURNED	15-May-2023
GENERAL COMMENTS	No comments